# PredRNN: Recurrent Neural Networks for Predictive Learning using Spatiotemporal LSTMs

**Yunbo Wang**
School of Software
Tsinghua University
wangyb15@mails.tsinghua.edu.cn

**Mingsheng Long**[*]
School of Software
Tsinghua University
mingsheng@tsinghua.edu.cn

**Jianmin Wang**
School of Software
Tsinghua University
jimwang@tsinghua.edu.cn

**Zhifeng Gao**
School of Software
Tsinghua University
gzf16@mails.tsinghua.edu.cn

**Philip S. Yu**
School of Software
Tsinghua University
psyu@uic.edu

## Abstract

The predictive learning of spatiotemporal sequences aims to generate future images by learning from the historical frames, where spatial appearances and temporal variations are two crucial structures. This paper models these structures by presenting a predictive recurrent neural network (PredRNN). This architecture is enlightened by the idea that spatiotemporal predictive learning should memorize both spatial appearances and temporal variations in a unified memory pool. Concretely, memory states are no longer constrained inside each LSTM unit. Instead, they are allowed to zigzag in two directions: across stacked RNN layers vertically and through all RNN states horizontally. The core of this network is a new Spatiotemporal LSTM (ST-LSTM) unit that extracts and memorizes spatial and temporal representations simultaneously. PredRNN achieves the state-of-the-art prediction performance on three video prediction datasets and is a more general framework, that can be easily extended to other predictive learning tasks by integrating with other architectures.

## 1 Introduction

As a key application of predictive learning, generating images conditioned on given consecutive frames has received growing interests in machine learning and computer vision communities. To learn representations of spatiotemporal sequences, recurrent neural networks (RNN) [17, 27] with the Long Short-Term Memory (LSTM) [9] have been recently extended from supervised sequence learning tasks, such as machine translation [22, 2], speech recognition [8], action recognition [28, 5] and video captioning [5], to this spatiotemporal predictive learning scenario [21, 16, 19, 6, 25, 12].

### 1.1 Why spatiotemporal memory?

In spatiotemporal predictive learning, there are two crucial aspects: spatial correlations and temporal dynamics. The performance of a prediction system depends on whether it is able to memorize relevant structures. However, to the best of our knowledge, the state-of-the-art RNN/LSTM predictive learning methods [19, 21, 6, 12, 25] focus more on modeling temporal variations (such as the object moving trajectories), with memory states being updated repeatedly over time inside each LSTM unit. Admittedly, the stacked LSTM architecture is proved powerful for supervised spatiotemporal learning

---

[*]Corresponding author: Mingsheng Long

(such as video action recognition [5, 28]). Two conditions are met in this scenario: (1) Temporal features are strong enough for classification tasks. In contrast, fine-grained spatial appearances prove to be less significant; (2) There are no complex visual structures to be modeled in the expected outputs so that spatial representations can be highly abstracted. However, spatiotemporal predictive leaning does not satisfy these conditions. Here, spatial deformations and temporal dynamics are equally significant to generating future frames. A straightforward idea is that if we hope to foretell the future, we need to memorize as many historical details as possible. When we recall something happened before, we do not just recall object movements, but also recollect visual appearances from coarse to fine. Motivated by this, we present a new recurrent architecture called Predictive RNN (PredRNN), which allows memory states belonging to different LSTMs to interact across layers (in conventional RNNs, they are mutually independent). As the key component of PredRNN, we design a novel Spatiotemporal LSTM (ST-LSTM) unit. It models spatial and temporal representations in a unified memory cell and convey the memory both vertically across layers and horizontally over states. PredRNN achieves the state-of-the-art prediction results on three video datasets. It is a general and modular framework for predictive learning and is not limited to video prediction.

## 1.2 Related work

Recent advances in recurrent neural network models provide some useful insights on how to predict future visual sequences based on historical observations. Ranzato *et al.* [16] defined a RNN architecture inspired from language modeling, predicting the frames in a discrete space of patch clusters. Srivastava *et al.* [21] adapted the sequence to sequence LSTM framework. Shi *et al.* [19] extended this model to further extract visual representations by exploiting convolutions in both input-to-state and state-to-state transitions. This Convolutional LSTM (ConvLSTM) model has become a seminal work in this area. Subsequently, Finn *et al.* [6] constructed a network based on ConvLSTMs that predicts transformations on the input pixels for next-frame prediction. Lotter *et al.* [12] presented a deep predictive coding network where each ConvLSTM layer outputs a layer-specific prediction at each time step and produces an error term, which is then propagated laterally and vertically in the network. However, in their settings, the predicted next frame always bases on the whole previous ground truth sequence. By contrast, we predict sequence from sequence, which is obviously more challenging. Patraucean *et al.* [15] and Villegas *et al.* [25] brought optical flow into RNNs to model short-term temporal dynamics, which is inspired by the two-stream CNNs [20] designed for action recognition. However, the optical flow images are hard to use since they would bring in high additional computational costs and reduce the prediction efficiency. Kalchbrenner *et al.* [10] proposed a Video Pixel Network (VPN) that estimates the discrete joint distribution of the raw pixel values in a video using the well-established PixelCNNs [24]. But it suffers from high computational complexity. Besides the above RNN architectures, other deep architectures are involved to solve the visual predictive learning problem. Oh *et al.* [14] defined a CNN-based action conditional autoencoder model to predict next frames in Atari games. Mathieu *et al.* [13] successfully employed generative adversarial networks [7, 4] to preserve the sharpness of the predicted frames.

In summary, these existing visual prediction models yield different shortcomings due to different causes. The RNN-based architectures [21, 16, 19, 6, 25, 12] model temporal structures with LSTMs, but their predicted images tend to blur due to a loss of fine-grained visual appearances. In contrast, CNN-based networks [13, 14] predict one frame at a time and generate future images recursively, which are prone to focus on spatial appearances and relatively weak in capturing long-term motions. In this paper, we explore a new RNN framework for predictive learning and present a novel LSTM unit for memorizing spatiotemporal information simultaneously.

## 2 Preliminaries

### 2.1 Spatiotemporal predictive learning

Suppose we are monitoring a dynamical system (e.g. a video clip) of $P$ measurements over time, where each measurement (e.g. a RGB channel) is recorded at all locations in a spatial region represented by an $M \times N$ grid (e.g. video frames). From the spatial view, the observation of these $P$ measurements at any time can be represented by a tensor $\mathcal{X} \in \mathbb{R}^{P \times M \times N}$. From the temporal view, the observations over $T$ time steps form a sequence of tensors $\mathcal{X}_1, \mathcal{X}_2, \ldots, \mathcal{X}_T$. The spatiotemporal predictive learning problem is to predict the most probable length-$K$ sequence in the future given the

previous length-$J$ sequence including the current observation:

$$\widehat{\mathcal{X}}_{t+1}, \ldots, \widehat{\mathcal{X}}_{t+K} = \underset{\mathcal{X}_{t+1}, \ldots, \mathcal{X}_{t+K}}{\arg\max} \; p\left(\mathcal{X}_{t+1}, \ldots, \mathcal{X}_{t+K} | \mathcal{X}_{t-J+1}, \ldots, \mathcal{X}_t\right). \tag{1}$$

Spatiotemporal predictive learning is an important problem, which could find crucial and high-impact applications in various domains: video prediction and surveillance, meteorological and environmental forecasting, energy and smart grid management, economics and finance prediction, etc. Taking video prediction as an example, the measurements are the three RGB channels, and the observation at each time step is a 3D video frame of RGB image. Another example is radar-based precipitation forecasting, where the measurement is radar echo values and the observation at every time step is a 2D radar echo map that can be visualized as an RGB image.

## 2.2 Convolutional LSTM

Compared with standard LSTMs, the convolutional LSTM (ConvLSTM) [19] is able to model the spatiotemporal structures simultaneously by explicitly encoding the spatial information into tensors, overcoming the limitation of vector-variate representations in standard LSTM where the spatial information is lost. In ConvLSTM, all the inputs $\mathcal{X}_1, \ldots, \mathcal{X}_t$, cell outputs $\mathcal{C}_1, \ldots, \mathcal{C}_t$, hidden state $\mathcal{H}_1, \ldots, \mathcal{H}_t$, and gates $i_t, f_t, g_t, o_t$ are 3D tensors in $\mathbb{R}^{P \times M \times N}$, where the first dimension is either the number of measurement (for inputs) or the number of feature maps (for intermediate representations), and the last two dimensions are spatial dimensions ($M$ rows and $N$ columns). To get a better picture of the inputs and states, we may imagine them as vectors standing on a spatial grid. ConvLSTM determines the future state of a certain cell in the $M \times N$ grid by the inputs and past states of its local neighbors. This can easily be achieved by using convolution operators in the state-to-state and input-to-state transitions. The key equations of ConvLSTM are shown as follows:

$$\begin{aligned}
g_t &= \tanh(\mathcal{W}_{xg} * \mathcal{X}_t + \mathcal{W}_{hg} * \mathcal{H}_{t-1} + b_g) \\
i_t &= \sigma(\mathcal{W}_{xi} * \mathcal{X}_t + \mathcal{W}_{hi} * \mathcal{H}_{t-1} + \mathcal{W}_{ci} \odot \mathcal{C}_{t-1} + b_i) \\
f_t &= \sigma(\mathcal{W}_{xf} * \mathcal{X}_t + \mathcal{W}_{hf} * \mathcal{H}_{t-1} + \mathcal{W}_{cf} \odot \mathcal{C}_{t-1} + b_f) \\
\mathcal{C}_t &= f_t \odot \mathcal{C}_{t-1} + i_t \odot g_t \\
o_t &= \sigma(\mathcal{W}_{xo} * \mathcal{X}_t + \mathcal{W}_{ho} * \mathcal{H}_{t-1} + \mathcal{W}_{co} \odot \mathcal{C}_t + b_o) \\
\mathcal{H}_t &= o_t \odot \tanh(\mathcal{C}_t),
\end{aligned} \tag{2}$$

where $\sigma$ is sigmoid activation function, $*$ and $\odot$ denote the convolution operator and the Hadamard product respectively. If the states are viewed as the hidden representations of moving objects, then a ConvLSTM with a larger transitional kernel should be able to capture faster motions while one with a smaller kernel can capture slower motions [19]. The use of the input gate $i_t$, forget gate $f_t$, output gate $o_t$, and input-modulation gate $g_t$ controls information flow across the memory cell $\mathcal{C}_t$. In this way, the gradient will be prevented from vanishing quickly by being trapped in the memory.

The ConvLSTM network adopts the encoder-decoder RNN architecture that is proposed in [23] and extended to video prediction in [21]. For a 4-layer ConvLSTM encoder-decoder network, input frames are fed into the the first layer and future video sequence is generated at the fourth one. In this process, spatial representations are encoded layer-by-layer, with hidden states being delivered from bottom to top. However, the memory cells that belong to these four layers are mutually independent and updated merely in time domain. Under these circumstances, the bottom layer would totally ignore what had been memorized by the top layer at the previous time step. Overcoming these drawbacks of this layer-independent memory mechanism is important to the predictive learning of video sequences.

## 3 PredRNN

In this section, we give detailed descriptions of the predictive recurrent neural network (PredRNN). Initially, this architecture is enlightened by the idea that a predictive learning system should memorize both spatial appearances and temporal variations in a unified memory pool. By doing this, we make the memory states flow through the whole network along a zigzag direction. Then, we would like to go a step further to see how to make the spatiotemporal memory interact with the original long short-term memory. Thus we make explorations on the memory cell, memory gate and memory fusion mechanisms inside LSTMs/ConvLSTMs. We finally derive a novel Spatiotemporal LSTM (ST-LSTM) unit for PredRNN, which is able to deliver memory states both vertically and horizontally.

## 3.1 Spatiotemporal memory flow

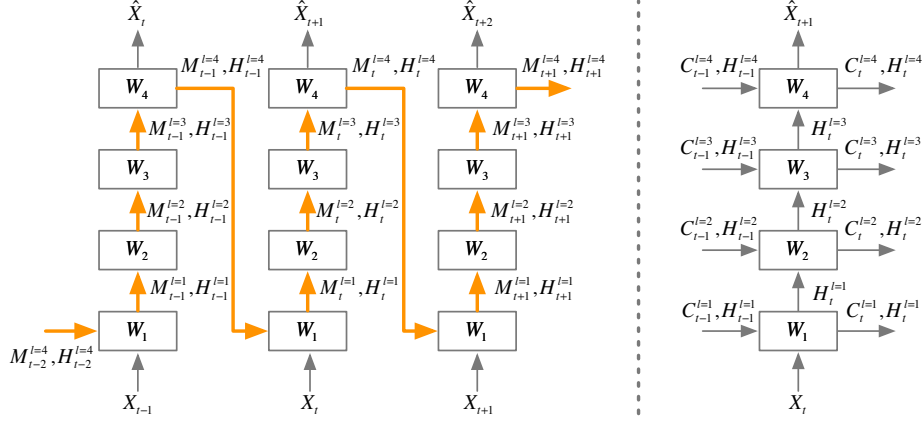

Figure 1: Left: The convolutional LSTM network with a spatiotemporal memory flow. Right: The conventional ConvLSTM architecture. The orange arrows denote the memory flow direction for all memory cells.

For generating spatiotemporal predictions, we initially exploit convolutional LSTMs (ConvLSTM) [19] as basic building blocks. Stacked ConvLSTMs extract highly abstract features layer-by-layer and then make predictions by mapping them back to the pixel value space. In the conventional ConvLSTM architecture, as illustrated in Figure 1 (right), the cell states are constrained inside each ConvLSTM layer and be updated only horizontally. Information is conveyed upwards only by hidden states. Such a temporal memory flow is reasonable in supervised learning, because according to the study of the stacked convolutional layers, the hidden representations can be more and more abstract and class-specific from the bottom layer upwards. However, we suppose in predictive learning, detailed information in raw input sequence should be maintained. If we want to see into the future, we need to learn from representations extracted at different-level convolutional layers. Thus, we apply a unified spatiotemporal memory pool and alter RNN connections as illustrated in Figure 1 (left). The orange arrows denote the feed-forward directions of LSTM memory cells. In the left figure, a unified memory is shared by all LSTMs which is updated along a zigzag direction. The key equations of the convolutional LSTM unit with a spatiotemporal memory flow are shown as follows:

$$
\begin{aligned}
g_t &= \tanh(\mathcal{W}_{xg} * \mathcal{X}_t \mathbb{1}_{\{l=1\}} + \mathcal{W}_{hg} * \mathcal{H}_t^{l-1} + b_g) \\
i_t &= \sigma(\mathcal{W}_{xi} * \mathcal{X}_t \mathbb{1}_{\{l=1\}} + \mathcal{W}_{hi} * \mathcal{H}_t^{l-1} + \mathcal{W}_{mi} \odot \mathcal{M}_t^{l-1} + b_i) \\
f_t &= \sigma(\mathcal{W}_{xf} * \mathcal{X}_t \mathbb{1}_{\{l=1\}} + \mathcal{W}_{hf} * \mathcal{H}_t^{l-1} + \mathcal{W}_{mf} \odot \mathcal{M}_t^{l-1} + b_f) \\
\mathcal{M}_t^l &= f_t \odot \mathcal{M}_t^{l-1} + i_t \odot g_t \\
o_t &= \sigma(\mathcal{W}_{xo} * \mathcal{X}_t \mathbb{1}_{\{l=1\}} + \mathcal{W}_{ho} * \mathcal{H}_t^{l-1} + \mathcal{W}_{mo} \odot \mathcal{M}_t^l + b_o) \\
\mathcal{H}_t^l &= o_t \odot \tanh(\mathcal{M}_t^l).
\end{aligned}
\tag{3}
$$

The input gate, input modulation gate, forget gate and output gate no longer depend on the hidden states and cell states from the previous time step at the same layer. Instead, as illustrated in Figure 1 (left), they rely on hidden states $\mathcal{H}_t^{l-1}$ and cell states $\mathcal{M}_t^{l-1}(l \in \{1, ..., L\})$ that are updated by the previous layer at current time step. Specifically, the bottom LSTM unit receives state values from the top layer at the previous time step: for $l = 1$, $\mathcal{H}_t^{l-1} = \mathcal{H}_{t-1}^L$, $\mathcal{M}_t^{l-1} = \mathcal{M}_{t-1}^L$. The four layers in this figure have different sets of input-to-state and state-to-state convolutional parameters, while they maintain a spatiotemporal memory cell and update its states separately and repeatedly as the information flows through the current node. Note that in the revised ConvLSTM network with a spatiotemporal memory in Figure 1, we replace the notation for memory cell from $\mathcal{C}$ to $\mathcal{M}$ to emphasize that it flows in the zigzag direction instead of the horizontal direction.

## 3.2 Spatiotemporal LSTM

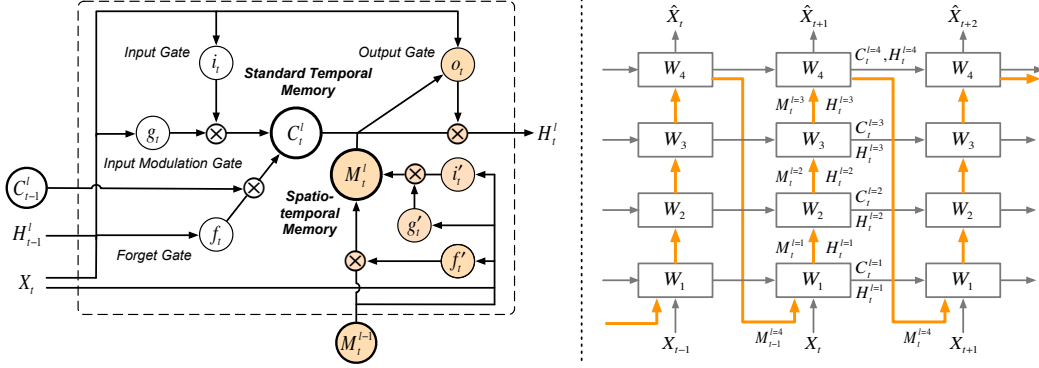

Figure 2: ST-LSTM (left) and PredRNN (right). The orange circles in the ST-LSTM unit denotes the differences compared with the conventional ConvLSTM. The orange arrows in PredRNN denote the spatiotemporal memory flow, namely the transition path of spatiotemporal memory $\mathcal{M}_t^l$ in the left.

However, dropping the temporal flow in the horizontal direction is prone to sacrificing temporal coherency. In this section, we present the predictive recurrent neural network (PredRNN), by replacing convolutional LSTMs with a novel spatiotemporal long short-term memory (ST-LSTM) unit (see Figure 2). In the architecture presented in the previous subsection, the spatiotemporal memory cells are updated in a zigzag direction, and information is delivered first upwards across layers then forwards over time. This enables efficient flow of spatial information, but is prone to vanishing gradient since the memory needs to flow a longer path between distant states. With the aid of ST-LSTMs, our PredRNN model in Figure 2 enables simultaneous flows of both standard temporal memory and the proposed spatiotemporal memory. The equations of ST-LSTM are shown as follows:

$$
\begin{aligned}
g_t &= \tanh(\mathcal{W}_{xg} * \mathcal{X}_t + \mathcal{W}_{hg} * \mathcal{H}_{t-1}^l + b_g) \\
i_t &= \sigma(\mathcal{W}_{xi} * \mathcal{X}_t + \mathcal{W}_{hi} * \mathcal{H}_{t-1}^l + b_i) \\
f_t &= \sigma(\mathcal{W}_{xf} * \mathcal{X}_t + \mathcal{W}_{hf} * \mathcal{H}_{t-1}^l + b_f) \\
\mathcal{C}_t^l &= f_t \odot \mathcal{C}_{t-1}^l + i_t \odot g_t \\
g_t' &= \tanh(\mathcal{W}_{xg}' * \mathcal{X}_t + \mathcal{W}_{mg} * \mathcal{M}_t^{l-1} + b_g') \\
i_t' &= \sigma(\mathcal{W}_{xi}' * \mathcal{X}_t + \mathcal{W}_{mi} * \mathcal{M}_t^{l-1} + b_i') \\
f_t' &= \sigma(\mathcal{W}_{xf}' * \mathcal{X}_t + \mathcal{W}_{mf} * \mathcal{M}_t^{l-1} + b_f') \\
\mathcal{M}_t^l &= f_t' \odot \mathcal{M}_t^{l-1} + i_t' \odot g_t' \\
o_t &= \sigma(\mathcal{W}_{xo} * \mathcal{X}_t + \mathcal{W}_{ho} * \mathcal{H}_{t-1}^l + \mathcal{W}_{co} * \mathcal{C}_t^l + \mathcal{W}_{mo} * \mathcal{M}_t^l + b_o) \\
\mathcal{H}_t^l &= o_t \odot \tanh(\mathcal{W}_{1\times1} * [\mathcal{C}_t^l, \mathcal{M}_t^l]).
\end{aligned}
\tag{4}
$$

Two memory cells are maintained: $\mathcal{C}_t^l$ is the standard temporal cell that is delivered from the previous node at $t-1$ to the current time step within each LSTM unit. $\mathcal{M}_t^l$ is the spatiotemporal memory we described in the current section, which is conveyed vertically from the $l-1$ layer to the current node at the same time step. For the bottom ST-LSTM layer where $l=1$, $\mathcal{M}_t^{l-1} = \mathcal{M}_{t-1}^L$, as described in the previous subsection. We construct another set of gate structures for $\mathcal{M}_t^l$, while maintaining the original gates for $\mathcal{C}_t^l$ in standard LSTMs. At last, the final hidden states of this node rely on the fused spatiotemporal memory. We concatenate these memory derived from different directions together and then apply a $1 \times 1$ convolution layer for dimension reduction, which makes the hidden state $\mathcal{H}_t^l$ of the same dimensions as the memory cells. Different from simple memory concatenation, the ST-LSTM unit uses a shared output gate for both memory types to enable seamless memory fusion, which can effectively model the shape deformations and motion trajectories in the spatiotemporal sequences.

# 4  Experiments

Our model is demonstrated to achieve the state-of-the-art performance on three video prediction datasets including both synthetic and natural video sequences. Our PredRNN model is optimized with a $L1 + L2$ loss (other losses have been tried, but $L1 + L2$ loss works best). All models are trained using the ADAM optimizer [11] with a starting learning rate of $10^{-3}$. The training process is stopped after $80,000$ iterations. Unless otherwise specified, the batch size of each iteration is set to 8. All experiments are implemented in TensorFlow [1] and conducted on NVIDIA TITAN-X GPUs.

## 4.1  Moving MNIST dataset

**Implementation**  We generate Moving MNIST sequences with the method described in [21]. Each sequence consists of 20 consecutive frames, 10 for the input and 10 for the prediction. Each frame contains two or three handwritten digits bouncing inside a $64 \times 64$ grid of image. The digits were chosen randomly from the MNIST training set and placed initially at random locations. For each digit, we assign a velocity whose direction is randomly chosen by a uniform distribution on a unit circle, and whose amplitude is chosen randomly in $[3, 5)$. The digits bounce-off the edges of image and occlude each other when reaching the same location. These properties make it hard for a model to give accurate predictions without learning the inner dynamics of the movement. With digits generated quickly on the fly, we are able to have infinite samples size in the training set. The test set is fixed, consisting of 5,000 sequences. We sample digits from the MNIST test set, assuring the trained model has never seen them before. Also, the model trained with two digits is tested on another Moving MNIST dataset with three digits. Such a test setup is able to measure PredRNN's generalization and transfer ability, because no frames containing three digits are given throughout the training period.

As a strong competitor, we include the latest state-of-the-art VPN model [10]. We find it hard to reproduce VPN's experimental results on Moving MNIST since it is not open source, thus we adopt its baseline version that uses CNNs instead of PixelCNNs as its decoder and generate each frame in one pass. We observe that the total number of hidden states has a strong impact on the final accuracy of PredRNN. After a number of trials, we present a 4-layer architecture with 128 hidden states in each layer, which yields a high prediction accuracy using reasonable training time and memory footprint.

Table 1: Results of PredRNN with spatiotemporal memory $\mathcal{M}$, PredRNN with ST-LSTMs, and state-of-the-art models. We report per-frame MSE and Cross-Entropy (CE) of generated sequences averaged across the Moving MNIST test sets. Lower MSE or CE denotes better prediction accuracy.

| Model | MNIST-2 (CE/frame) | MNIST-2 (MSE/frame) | MNIST-3 (MSE/frame) |
|---|---|---|---|
| FC-LSTM [21] | 483.2 | 118.3 | 162.4 |
| ConvLSTM ($128 \times 4$) [19] | 367.0 | 103.3 | 142.1 |
| CDNA [6] | 346.6 | 97.4 | 138.2 |
| DFN [3] | 285.2 | 89.0 | 130.5 |
| VPN baseline [10] | 110.1 | 70.0 | 125.2 |
| PredRNN with spatiotemporal memory $\mathcal{M}$ | 118.5 | 74.0 | 118.2 |
| **PredRNN + ST-LSTM** ($128 \times 4$) | 97.0 | **56.8** | **93.4** |

**Results**  As an ablation study, PredRNN only with a zigzag memory flow reduces the per-frame MSE to 74.0 on the Moving MNIST-2 test set (see Table 1). By replacing convolutional LSTMs with ST-LSTMs, we further decline the sequence MSE from 74.0 down to 56.8. The corresponding frame-by-frame quantitative comparisons are presented in Figure 3. Compared with VPN, our model turns out to be more accurate for long-term predictions, especially on Moving MNIST-3. We also use per-frame cross-entropy likelihood as another evaluation metric on Moving MNIST-2. PredRNN with ST-LSTMs significantly outperforms all previous methods, while PredRNN with spatiotemporal memory $\mathcal{M}$ performs comparably with VPN baseline.

A qualitative comparison of predicted video sequences is given in Figure 4. Though VPN's generated frames look a bit sharper, its predictions gradually deviate from the correct trajectories, as illustrated in the first example. Moreover, for those sequences that digits are overlapped and entangled, VPN has difficulties in separating these digits clearly while maintaining their individual shapes. For example,

in the right figure, digit "8" loses its left-side pixels and is predicted as "3" after overlapping. Other baseline models suffer from a severer blur effect, especially for longer future time steps. By contrast, PredRNN's results are not only sharp enough but also more accurate for long-term motion predictions.

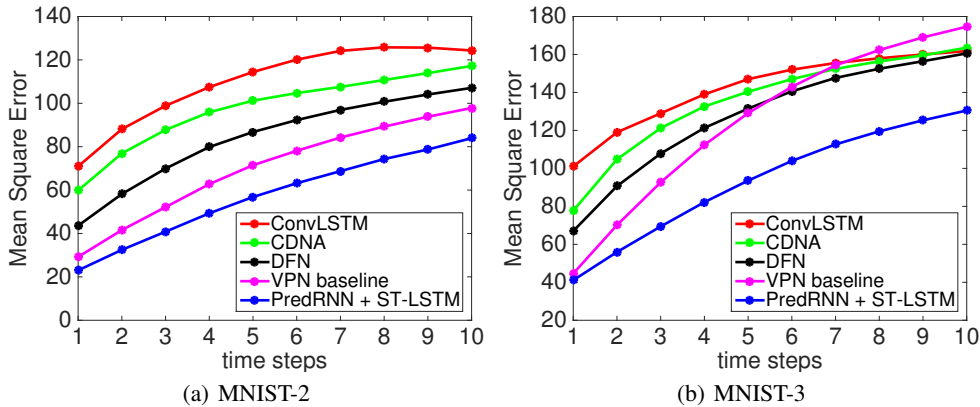

(a) MNIST-2                    (b) MNIST-3

Figure 3: Frame-wise MSE comparisons of different models on the Moving MNIST test sets.

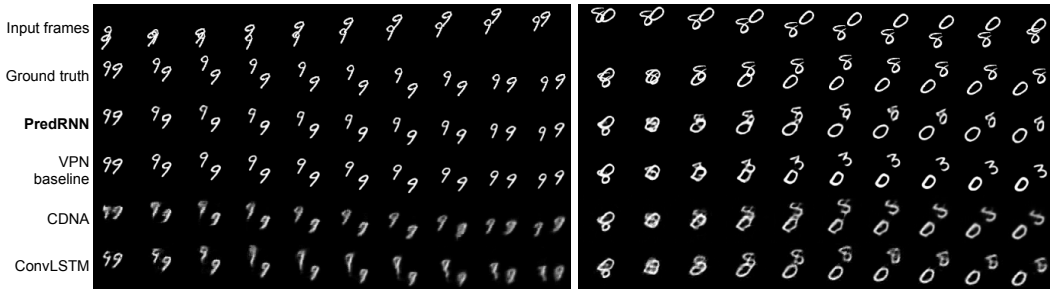

Figure 4: Prediction examples on the Moving MNIST-2 test set.

## 4.2 KTH action dataset

**Implementation** The KTH action dataset [18] contains six types of human actions (walking, jogging, running, boxing, hand waving and hand clapping) performed several times by 25 subjects in four different scenarios: outdoors, outdoors with scale variations, outdoors with different clothes and indoors. All video clips were taken over homogeneous backgrounds with a static camera in 25fps frame rate and have a length of four seconds in average. To make the results comparable, we adopt the experiment setup in [25] that video frames are resized into $128 \times 128$ pixels and all videos are divided with respect to the subjects into a training set (persons 1-16) and a test set (persons 17-25). All models, including PredRNN as well as the baselines, are trained on the training set across all six action categories by generating the subsequent 10 frames from the last 10 observations, while the the presented prediction results in Figure 5 and Figure 6 are obtained on the test set by predicting 20 time steps into the future. We sample sub-clips using a 20-frame-wide sliding window with a stride of 1 on the training set. As for evaluation, we broaden the sliding window to 30-frame-wide and set the stride to 3 for running and jogging, while 20 for the other categories. Sub-clips for running, jogging, and walking are manually trimmed to ensure humans are always present in the frame sequences. In the end, we split the database into a training set of 108,717 sequences and a test set of 4,086 sequences.

**Results** We use the Peak Signal to Noise Ratio (PSNR) and the Structural Similarity Index Measure (SSIM) [26] as metrics to evaluate the prediction results and provide frame-wise quantitative comparisons in Figure 5. A higher value denotes a better prediction performance. The value of SSIM ranges between -1 and 1, and a larger score means a greater similarity between two images. PredRNN consistently outperforms the comparison models. Specifically, the Predictive Coding Network [12] always exploits the whole ground truth sequence before the current time step to predict the next

frame. Thus, it cannot make sequence predictions. Here, we make it predict the next 20 frames by feeding the 10 ground truth frames and the recursively generated frames in all previous time steps. The performance of MCnet [25] deteriorates quickly for long-term predictions. Residual connections of MCnet convey the CNN features of the last frame to the decoder and ignore the previous frames, which emphasizes the spatial appearances while weakens temporal variations. By contrast, results of PredRNN in both metrics remain stable over time, only with a slow and reasonable decline. Figure 6 visualizes a sample video sequence from the KTH test set. The ConvLSTM network [19] generates blurred future frames, since it fails to memorize the detailed spatial representations. MCnet [25] produces sharper images but is not able to forecast the movement trajectory accurately. Thanks to ST-LSTMs, PredRNN memorizes detailed visual appearances as well as long-term motions. It outperforms all baseline models and shows superior predicting power both spatially and temporally.

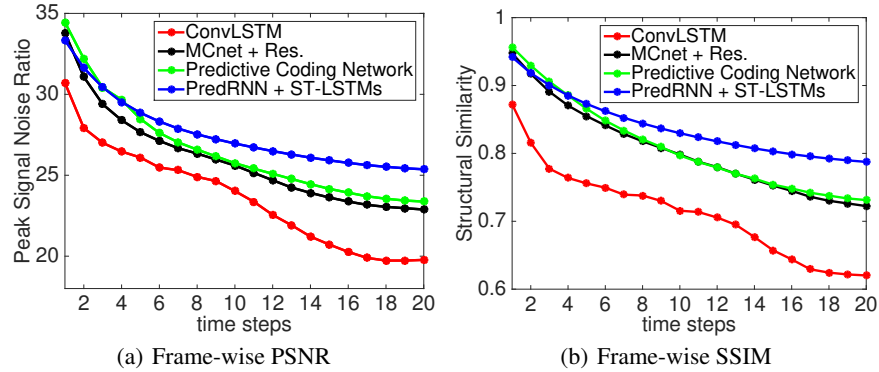

(a) Frame-wise PSNR　　　　　　　　(b) Frame-wise SSIM

Figure 5: Frame-wise PSNR and SSIM comparisons of different models on the KTH action test set. A higher score denotes a better prediction accuracy.

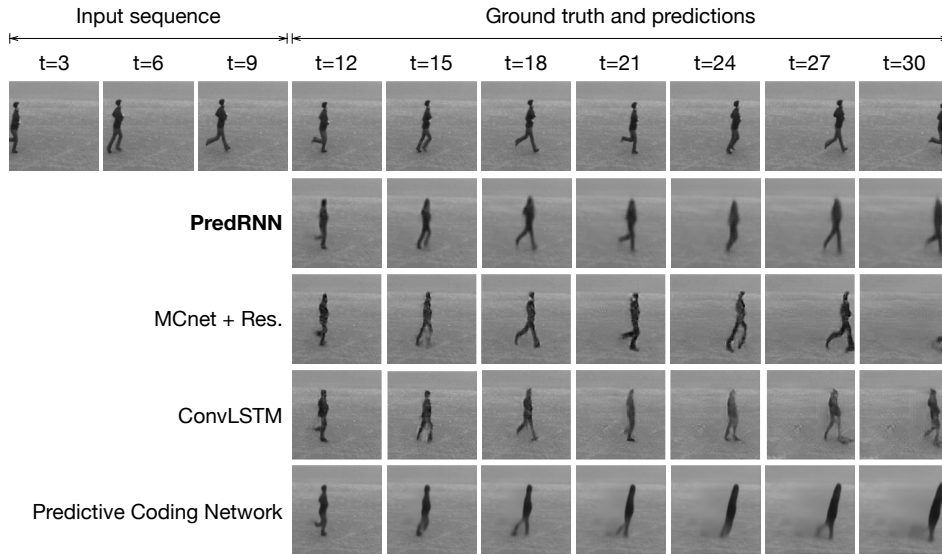

Figure 6: KTH prediction samples. We predict 20 frames into the future by observing 10 frames.

### 4.3 Radar echo dataset

Predicting the shape and movement of future radar echoes is a real application of predictive learning and is the foundation of precipitation nowcasting. It is a more challenging task because radar echoes are not rigid. Also, their speeds are not as fixed as moving digits, their trajectories are not as periodical as KTH actions, and their shapes may accumulate, dissipate or change rapidly due to the complex atmospheric environment. Modeling spatial deformation is significant for the prediction of this data.

**Implementation** We first collect the radar echo dataset by adapting the data handling method described in [19]. Our dataset consists of 10,000 consecutive radar observations, recorded every 6 minutes in Guangzhou, China. For preprocessing, we first map the radar intensities to pixel values, and represent them as $100 \times 100$ gray-scale images. Then we slice the consecutive images with a 20-frame-wide sliding window. Thus, each sequence consists of 20 frames, 10 for the input, and 10 for forecasting. The total 9,600 sequences are split into a training set of 7,800 samples and a test set of 1,800 samples. The PredRNN model consists of two ST-LSTM layers with 128 hidden states each. The convolution filters inside ST-LSTMs are set to $3 \times 3$. After prediction, we transform the resulted echo intensities into colored radar maps, as shown in Figure 7, and then calculate the amount of precipitation at each grid cell of these radar maps using Z-R relationships. Since it would bring in an additional systematic error to rainfall prediction and makes final results misleading, we do not take them into account in this paper, but only compare the predicted echo intensity with the ground truth.

**Results** Two baseline models are considered. The ConvLSTM network [19] is the first architecture that models sequential radar maps with convolutional LSTMs, but its predictions tend to blur and obviously inaccurate (see Figure 7). As a strong competitor, we also include the latest state-of-the-art VPN model [10]. The PixelCNN-based VPN predicts an image pixel by pixel recursively, which takes around 15 minutes to generate a radar map. Given that precipitation nowcasting has a high demand on real-time computing, we trade off both prediction accuracy and computation efficiency and adopt VPN's baseline model that uses CNNs as its decoders and generates each frame in one pass. Table 2 shows that the prediction error of PredRNN is significantly lower than VPN baseline. Though VPN generates more accurate radar maps for the near future, it suffers from a rapid decay for the long term. Such a phenomenon results from a lack of strong LSTM layers to model spatiotemporal variations. Furthermore, PredRNN takes only 1/5 memory space and training time as VPN baseline.

Table 2: Quantitative results of different methods on the radar echo dataset.

| Model | MSE/frame | Training time/100 batches | Memory usage |
|---|---|---|---|
| ConvLSTM [19] | 68.0 | 105 s | 1756 MB |
| VPN baseline [10] | 60.7 | 539 s | 11513 MB |
| **PredRNN** | **44.2** | **117 s** | **2367 MB** |

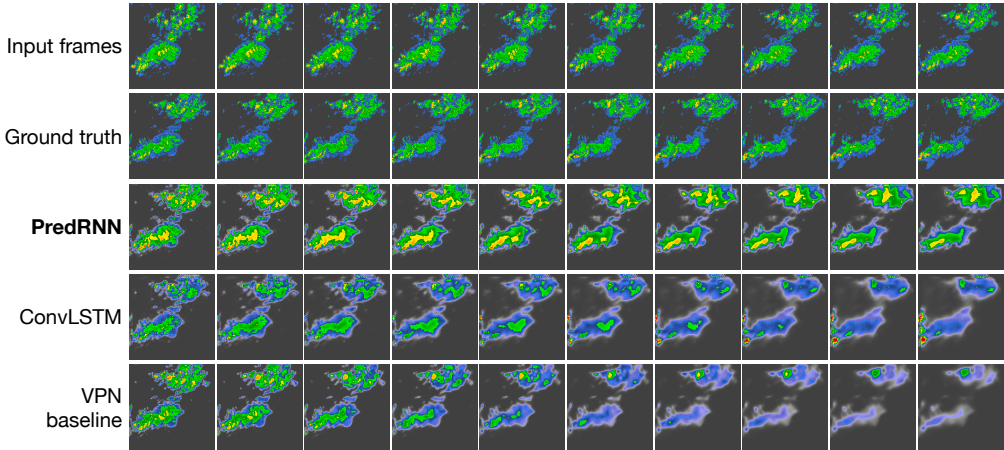

Figure 7: A prediction example on the radar echo test set.

## 5  Conclusions

In this paper, we propose a novel end-to-end recurrent network named PredRNN for spatiotemporal predictive learning that models spatial deformations and temporal variations simultaneously. Memory states zigzag across stacked LSTM layers vertically and through all time states horizontally. Furthermore, we introduce a new spatiotemporal LSTM (ST-LSTM) unit with a gate-controlled dual memory structure as the key building block of PredRNN. Our model achieves the state-of-the-art performance on three video prediction datasets including both synthetic and natural video sequences.

## Acknowledgments

This work was supported by the National Key R&D Program of China (2016YFB1000701), National Natural Science Foundation of China (61772299, 61325008, 61502265, 61672313) and TNList Fund.

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
