[Reviews · NeurIPS 2017]

Reviewer 1



The work introduces a new architecture for conditional video generation. It is is based heavily on convolutional LSTMs, with the insight that the prior architecture where each layer of hierarchy represents increasingly abstract properties of the scene may be suboptimal for video generation (unlike classification, a generative model needs, even at the output layer, to retain information about the precise position of objects in the scene). The model introduced here, PredRNN extends convolutional LSTMs to contain two memory cells, one which flows through time at the same layer (like the original ConvLSTM), and one which flows up through the layers. The authors test the model on two datasets: moving MNIST digits and KTH action recognition dataset and demonstrate superior MSE in video prediction to prior work. The paper is fairly well-written and cites prior work. The architecture is novel and interesting. I think the comparison with prior work could be strengthened which would make it easier to understand the empirical strength of these results. For example, the Kalchbrenner, 2016 claims to reach near the lower-bound the MNIST digit task, but is not included as a baseline. Prior work on MNIST (e.g. (Kalchbrenner, 2016) and the original introduction of the dataset, (Srivastava, 2016) report the cross-entropy loss. Here, the authors focus on the maximum likelihood output, but it would be helpful for comparison with prior work to also report the likelihood. Additionally, the computational complexity is mentioned as an advantage of this model, but no detailed analysis or comparison is performed so its hard to know how this compares computational complexity with prior work. Minor notational suggestion: It might be easier for the reader to follow if you use M instead of C for the cell state in equation 3 so that the connection with equation 4 is clearer. [I've read the author's response. I think this paper is stronger for comparison with prior work (which I assume they'll include in the final version) so I have raised my evaluation. I'm still unclear if they are training with MSE and other approaches are using different losses, doesn't that provide an advantage to this model when evaluating using MSE?]

Reviewer 2



This paper deals with predictive learning (mainly video prediction), using a RNN type of structure. A new (to my knowledge) variation of LSTM is introduced, called ST-LSTM, with recurrent connections not only in the forward time direction. The predictive network is composed of ST-LSTM blocks. Each of these block resemble a convolutional LSTM unit, with some differences to include an additional input. This extra input comes from the last layer of the previous time step, and enters at the first layer of the current time step, it is then propagated through the layers at the current time step. The resulting ST-LSTM is similar to the combination of two independent LSTM units, interacting through concatenation of the memories (figure 2(left)). I couldn't find an explicit formulation of the loss (the closest I could find is equation 1). I am assuming that it is a MSE loss, but this should be confirmed, since other forms are possible. Results are presented on the Moving MNIST (synthetic) and KTH (natural) datasets, showing both PSNR/SSIM and generations. The authors compare their PredRNN with other baselines, and show the best results on these datasets. On Moving MNIST, there is a comparison of different LSTM schemes, and ST-LSTM show the best results, which is a good result. However, the experimental section could be stronger by having more datasets (the two datasets presented have little ambiguity in their future, it could be interesting to see how the model preforms in less contained dataset, such as Sports1m, UCF101 or the Google "Push" dataset, for instance). Although this paper present a complex (and, as it seems, good) structure for the predictor, the loss function is almost never mentioned. As (among others) [17] and [19] mention, a simple loss such as the MSE cannot work well when the future distribution is multimodal. The paper would also be stronger if compared with other methods (non LSTM-based), such as the ones presented in section 1.2. In particular, VPNs and GANs seem like strong competitors. Notes: - Figure 1: Although the figure is clear, I do not quite understand what the orange arrows mean (compared to the black arrows). - Figure 2(right): As I understand it, each W box should also have an Xt input (not just W1)

Reviewer 3



This paper introduces a novel convolutional LSTM based architecture for next frame video prediction. The difference with previous attempts is that spatial and temporal variations are gathered in a single memory pool. Comments The paper is mostly well written, proposes a new approach that appears to be fruitful on two relatively simple datasets. Providing generated videos for such paper submission would be appreciated. The results of the paper seem good but the evaluation of the proposed approach to real natural images would be more convincing. The KTH dataset is described as a dataset of "natural images sequences" but remain very simple to analyse: very low resolution, uniform foreground and background... As the proposed approach is claimed to be memory efficient, it shouldn't be a problem. Could you provide an idea of the training time? l.84-86 the authors describe the applications of video prediction as a fact in numerous domains, without citing any references. The reader is therefore curious if these applications are already happening in this case, the authors should provide references, or might happen later, in this case the authors should rephrase (finds -> could find) l. 105-106 I don't understand the introduction of the deconvolution operator, since it seems unused in the equations and the rest of the paper. Minor: l.47 no comma l 57-58 they ... always -> they always l 57 the next one -> the next l. 135 no comma l 141 in in -> in l 135-137 I don't understand this sentence l 154 : We -> we l 242: ignores -> ignore [9] {LSTM} ICML 15 [15] ICLR workshop 16 [19]: ICLR 16 [23] {SVM} [24] ICLR 15 please check other references